# Roles of miR-196a and miR-196b in Zebrafish Motor Function

**DOI:** 10.3390/biom13030554

**Published:** 2023-03-17

**Authors:** Chunyan Yuan, Huaping Xie, Xiangding Chen, Shunling Yuan

**Affiliations:** 1Laboratory of Molecular and Statistical Genetics, College of Life Sciences, Hunan Normal University, Changsha 410081, China; 2582@hutb.edu.cn; 2College of Physical Education and Health, Hunan University of Technology and Business, Changsha 410025, China; 3Hunan Provincial Key Laboratory of Physical Fitness and Sports Rehabilitation, Hunan Normal University, Changsha 410012, China

**Keywords:** miR-196a-1, miR-196b, motor function, bone, muscle

## Abstract

Background: The exertion of motor function depends on various tissues, such as bones and muscles. miR-196 has been widely studied in cancer and other fields, but its effect on bone and skeletal muscle is rarely reported. In order to explore the role of miR-196 family in bone and skeletal muscle, we used the previously successfully constructed miR-196a-1 and miR-196b gene knockout zebrafish animal models for research. Methods: The behavioral trajectories of zebrafish from 4 days post-fertilization (dpf) to 7 dpf were detected to analyze the effect of miR-196a-1 and miR-196b on motor ability. Hematoxylin-eosin (HE) staining and transmission electron microscopy (TEM) were used to detect the dorsal muscle tissue of zebrafish. The bone tissue of zebrafish was detected by microcomputed tomography (micro-CT). Real-time PCR was used to detect the expression levels of related genes, including vcp, dpm1, acta1b, mylpfb, col1a1a, bmp8a, gdf6a, and fgfr3. Results: The behavioral test showed that the total behavioral trajectory, movement time, and movement speed of zebrafish larvae were decreased in the miR-196a-1 and miR-196b gene knockout lines. Muscle tissue analysis showed that the structure of muscle fibers in the zebrafish lacking miR-196a-1 and miR-196b was abnormal and was characterized by vacuolar degeneration of muscle fibers, intranuclear migration, melanin deposition, and inflammatory cell infiltration. Bone CT examination revealed decreased bone mineral density and trabecular bone number. The real-time PCR results showed that the expression levels of vcp, dpm1, gdf6a, fgfr3, and col1a1a were decreased in the miR-196b gene knockout group. The expression levels of dpm1, acta1b, mylpfb, gdf6a, and col1a1a were decreased, and the expression level of fgfr3 was increased in the miR-196b gene knockout group compared with the wild-type group. Conclusions: miR-196a-1 and miR-196b play an important role in muscle fiber structure, bone mineral density, and bone trabecular quantity by affecting the expression of vcp, dpm1, acta1b, mylpfb, gdf6a, fgfr3, and col1a1a and then affect the function of the motor system

## 1. Introduction

The motor system of the body is extremely complex, and its function depends on tissues such as bone and muscle [1,2,3]. Bone or skeletal muscle is needed to maintain the whole body, and abnormal function can cause an inability to properly exercise and lead to a variety of chronic diseases, such as osteoporosis, fractures, osseous arthritis, muscle atrophy, and issues with the lumbar disc. Diseases such as arthritis, gout, and weakening or loss of body movement seriously affect the daily life of an individual and their quality of life, resulting in a major economic burden to society and families [4,5,6]. In recent years, the incidence of motor system-related diseases has increased, and there is no effective treatment for many diseases. Therefore, it is important to study and further explore the role and regulatory mechanism of bone and muscle for the defense, treatment, recovery, and improvement of motor function in motor system diseases.

MicroRNAs (miRNAs) are a class of highly conserved noncoding small RNAs that are approximately 22 to 25 nucleotides in length, widely exist in a variety of eukaryotic cells and a large number of species, and play an important role in the growth, development, metabolism, regeneration, and physiological functions of bone and skeletal muscle [7,8,9,10,11].

Previous studies [12,13] have reported that miR-196 is related to bone mineral density and muscle. Studies regulating the expression level of miR-196 showed that this molecule plays an important role in preventing the occurrence of bone and skeletal muscle diseases and improving related diseases, suggesting that miR-196 may be involved in bone or skeletal muscle. miR-196 is encoded by the homologous HOX family, is highly conserved among different species, and includes miR-196a (mature sequence: 3′-GGG UUG UUG UAC UUU GAU GGA U-5′) and miR-196b (mature sequence: 3′-GGG UUG UUG UCC UUU GAU GGA U-5′), which are highly homologous with only one nucleotide difference [14,15].

A study on osteosarcoma reported that the expression level of miR-196a in osteosarcoma cells was significantly higher than that in normal tissues, and this molecule ultimately promoted the metastasis of tumor cells and extracellular matrix transformation by targeting the 3′-UTR of *HOXA5* mRNA [16]. Another study [17] on osteosarcoma reported that overexpressed miR-196a promotes cell proliferation and inhibits cell apoptosis through the PTEN/Akt/FOXO1 signaling pathway. Zhong et al. [13] found, in a mouse model of osteoporosis, that overexpression of miR-196a could inhibit the expression of the GNAS gene through the Hedgehog signaling pathway and promote osteogenic differentiation in mice. In addition, studies [18] have shown that miR-196a-5p inhibits the formation of osteoclast-like cells and mitochondrial energy metabolism in mouse cells. A miRNA microarray detection of osteosarcoma found that the expression of miR-196a and miR-196b was upregulated [19]. Another exosome experiment [20] examining the source of bone marrow stromal cells found that miR-196a played an important role in the differentiation of osteoblasts and the expression of related osteoblast genes. At present, there have been few studies on miR-196a and miR-196b in the bones and muscles of the motor system. In a study on the expression pattern of miR-196 in zebrafish, it was found that pri-miR-196a1 was expressed in the tail and trunk of 24 zebrafish embryos [21]. In summary, we speculate that miR-196a may play a role in bone or skeletal muscle.

In zebrafish, skeletal muscle and bone make up a large part of the body trunk and are highly similar to human muscle both molecularly and histologically, making them suitable for the study of bone and muscle diseases [22]. Additionally, the relatively simple genome and simple genetic manipulation of zebrafish are advantageous in myopathy studies [23].To explore the role of miR-196a-1 and miR-196b in the motor system, we constructed miR-196a and miR-196b gene knockout lines. In this study, we tested the behavior and motor function of miR-196a-1 and miR-196b gene knockout embryos, as well as muscle and bone tissue structure, to explore the influence of miR-196a-1 and miR-196b on the muscle of the operating system. The relevant results will further clarify the role of miR-196 in the body and contribute to its clinical application.

## 2. Materials and Methods

### 2.1. Zebrafish Lines

The miR-196a-1 and miR-196b gene knockout lines and the miR-196a-1 and miR-196b gene double knockout zebrafish lines were obtained for this experiment (College of Life Sciences, Hunan Normal University, Changsha, China). As we reported earlier, homozygous mutant zebrafish lines were constructed using CRISPR/Cas9 gene editing technology and were screened by gene sequencing [24]. Homozygous mutants of gene knockout zebrafish showed no abnormal embryonic development after breeding. Twelve zebrafish were randomly selected form each group, for a total of 48 zebrafish.

### 2.2. Zebrafish Maintenance

Zebrafish strains were raised in a water temperature of 28 °C and pH 6.5~7.4, with alternating cycles of 14 h light and 10 h dark in the Zebrafish Laboratory of the College of Life Sciences, Hunan Normal University. According to previous reports [25], the embryos were incubated at 28.5 °C water temperature, and paramecium (Heading, Tianjin, China) was given 5 days later. The young fish were transferred to a feeding system rack (ESEN, Beijing, China) after the 14th day and fed with heading (Tianjin, China) twice a day at a fixed time in the morning and evening.

### 2.3. Behavioral Test of the Zebrafish Model

The behavioral movement of zebrafish embryos at 4 dpf, 5 dpf, 6 dpf, and 7 dpf was detected in 24-well plates using the ViewPoint system (Lyon, France). Zebrafish embryos were cultured in 24-well cell culture plates filled with 60 μg/mL Instant Ocean salt mix with 1 larva per well. The total distance traveled by larvae at different speeds, the total duration of movement at different speeds, and the number of movements were recorded within 10 min.

### 2.4. HE Staining of Muscle Tissue

Zebrafish back muscle tissue was collected after feeding ten months, anhydrous ethanol dehydration (Characters, Shanghai, China) was performed, xylene (SINPHARM, Beijing, China) solution was added, and paraffin infiltration and the embedding of slices (Leica, Heidelberg, Germany) were performed. The muscle tissue was sliced after dewaxing, hydration and hematoxylin (St. Louis, MO, USA) and eosin (Solarbio, Beijing, China) dyeing, and finally sliced and photographed.

### 2.5. Transmission Electron Microscopy of Muscle

The back muscle tissue of ten-month-old zebrafish (1 mm^3^) was fixed in 4 °C 2.5% glutaraldehyde solution. The fixative solution was discarded, and cells were transferred to phosphate-buffered saline. Next, cells were fixed in 1% osmic acid for 1–2 h, and dehydration was carried out by incubation in 30% ethanol for 10 min, 50% ethanol for 10 min, 70% uranyl acetate in ethanol (stained before embedding) for 3 h or overnight, 80% ethanol for 10 min, 95% ethanol for 15 min, 100% ethanol twice for 50 min each, and propylene oxide for 30 min. Next, samples were incubated in propylene oxide: epoxy resin (1:1) for 1–2 h and then in pure epoxy resin for 2–3 h. After embedding in pure epoxy resin, samples were baked in an oven at 40 °C for 12 h and then at 60 °C for 48 h. Samples were then cut into ultra-thin sections and placed on copper grids. Staining was then performed with lead and uranium stain, and images were acquired using a Japan Electronics JEM1400 transmission electron microscope and recorded with a Morada G3 digital camera.

### 2.6. Bone Micro-CT Examination

Ten-month-old zebrafish were collected, fixed with 4% paraformaldehyde, and then sent to Xidian University (Xi’an, China) for micro-CT (microcomputed tomography) analysis. A hardware system (Xidian University, Xi’an, China) composed of an X-ray tube, rotary table, detector, and acquisition card was used for shooting detection. The analysis was carried out by the Hiscan Analyzer software system (Hiscan, Suzhou, China). According to the section image information obtained from the scan, graying, binarization, head image segmentation of the target area, and threshold segmentation were carried out sequentially; the influence of cortical bone (or cancellous bone), soft tissue, and fluid in the medullary cavity was excluded; and finally, statistical analyses were carried out.

### 2.7. Real-Time PCR Analysis of Skeletal Muscle and Skeletal-Related Gene Expression Levels

The dorsal muscle tissue and bone tissue of 10-year-old zebrafish were collected, total RNA was extracted from tissue, and cDNA was synthesized by reverse transcriptase (Thermo, MA, USA). Then, a SYBR Green qRCR kit (TaKaRa, Tokyo, Japan) and specific primers were used for amplification. The tested genes included vcp (R:CAGAGAAGAACGCACCAGCCATC, F:CCCTTTGCTTGAGTCCGTCCATC); *dpm1* (R:AGCCGGAGAAGTAATGCGAA, F:CGGGAGGTTTTCTCGCTCAT); *acta1b* (R:CTGGCACCACACCTTCTACAATGAG, F:GGTCATCTTCTCACGGTTAGCCTTG); *mylpfb* (R:GATGTGCTGGCAACAATGGG, F:GCGCCCTTTAGCTTTTCACC); *col1a1a* (R:GCAGCACTTCCAGCACCCTTAC, F:AGGAGCACCAGCAATACCAGGAG); *bmp8a* (R:ATGGACAGACACGAGGTTGAGATTG, F:TACACACAGAGGGAGGAAGATGGAC); *gdf6a* (R:ACCGTCTGGACAGGATTCACTAAGG, F:TCAACAGGTGCTCGTCTACACATTC); and *fgfr3* (R:AGATGAGGACGAGGCAGGTAATGG, F:CAGCAGGACAGCGGAACTTGAC). The cycle threshold (CT) value was collected, and the formula 2^−ΔΔC^ was used to calculate the relative expression level of target genes in each sample.

### 2.8. Statistical Analysis

Data between different groups are expressed as the mean ± variance. The experiment was repeated three times. SPSS 20.0 software (SPSS, Inc., Chicago, IL, USA) was used for statistical analysis. A two-tailed Student’s *t* test or one-way analysis of variance was used to compare the differences between different groups. *p* < 0.05 was considered statistically significant.

## 3. Results

### 3.1. Muscle Behavior Analysis of miR-196a-1 and miR-196b Gene Knockout Zebrafish

The motor behaviors and behavioral trajectories of zebrafish embryos on the fourth, fifth, sixth, and seventh days were recorded for 10 min every day, and the experiment was repeated three times. We captured and recorded the activity track of each group of zebrafish in a 24-hole culture plate and selected three holes for each group to display, as shown in Figure 1.

The behavior track and activity time of zebrafish larvae of the four groups in the culture plate were collected, and the total movement distance, total duration, movement speed, and movement times of the zebrafish in each group were calculated.

The behavior trajectories of zebrafish larvae were observed continuously for 4 days. According to the detection data, the total moving distance and moving times of both wild-type zebrafish and miR-196a-1 and miR-196b gene knockout zebrafish increased obviously with the development of larvae.

The behavioral data of the fourth, fifth, sixth, and seventh days were calculated, and we found that the total movement distance of the miR-196a-1 or miR-196b^−/−^ group (Figure 2A) was smaller than that of the wild-type zebrafish, and the differences were statistically significant (*n* = 24, *p* < 0.05). Compared with that of the miR-196a-1 or miR-196b knockout groups, the total travel distance of the miR-196a-1 and miR-196b double knockout groups was further reduced, but the difference between the groups was not statistically significant. The total movement time of zebrafish was statistically analyzed (Figure 2B). The total movement time of the miR-196a-1 gene knockout group, miR-196b gene knockout group, and double knockout group was decreased compared with that of the wild-type control group. In a comparison of the movement speed of each group (Figure 2C), the speed of the miR-196a-1 or miR-196b gene knockdown group decreased, but the difference between groups was not significant. The number of movements of different groups (Figure 2D) was determined, the number of movements of the miR-196a-1 or miR-196b gene knockdown group decreased, and further decreased after the double knockdown of miR-196a-1 and miR-196b; the differences between some groups were statistically significant.

In summary, the experimental results suggest that the total behavioral trajectory of zebrafish larvae is reduced, and the movement time and speed are also reduced after miR-196a-1 and miR-196b gene knockout, indicating that the motor ability is decreased. When both the miR-196a-1 and miR-196b genes were knocked out, the motor ability was further decreased compared with that of the miR-196a-1 or miR-196b single knockout groups. However, there was no significant difference between the miR-196a-1 and miR-196b gene knockdown groups. These results indicated that both the miR-196a-1 and miR-196b genes play a role in the motor ability of zebrafish.

### 3.2. HE Staining of Muscle Tissue of the miR-196a-1 and miR-196b Knockout Zebrafish Model

To further explore the influence of the miR-196a-1 and miR-196b genes on motor function, we assessed the muscle and bone tissues of zebrafish with gene knockout.

The dorsal muscle tissue of zebrafish from the wild-type, miR-196a-1 gene knockout, miR-196b gene knockout, and miR-196a-1 and miR-196b gene double knockout groups were stained with HE, and the results of longitudinal and transverse sections are shown in Figure 3.

The transverse section of the dorsal muscle tissue of wild-type zebrafish (Figure 3A1) showed that the muscle fibers were polyangular and uniform in size. The muscle nuclei were distributed around the muscle fibers, and the nuclear membrane was intact. In the longitudinal section of zebrafish in the wild-type group (Figure 3A2), muscle fibers were arranged neatly with uniform thickness, and a small amount of inflammatory cell infiltration was occasionally observed in the interstitium. In the transverse section and longitudinal section of the muscle in the miR-196a-1 knockout group (Figure 3B1), the tissue structure of the muscle fibers was slightly abnormal, some muscle fibers showed vacuolar degeneration, and a small amount of inflammatory cells could be seen in the intercellular substance. In the miR-196b knockout zebrafish group (Figure 3C), muscle fibers were slightly abnormal, vacuolar degeneration of some muscle fibers was present, and a small amount of inflammatory cells were observed in the transverse section of the muscle. A small number of inflammatory cells in the interstitial tissue and individual melanin deposition could be seen in the longitudinal muscle section (Figure 3C2). The tissue structure of muscle fibers was slightly abnormal in the miR-196a-1 and miR-196b double knockout groups (Figure 3D1). The shape of the muscle fibers was irregular, with enlarged and rounded muscle fibers, vacuolar degeneration of some muscle fibers, infiltration of a small amount of inflammatory cells in the stroma, individual melanin deposition in the tissue, and intranuclear migration of individual muscle cells. Longitudinal sections (Figure 3D2) showed that individual inflammatory cells infiltrated the interstitium, and individual muscle cells were hypertrophic. The results suggest that the absence of miR-196a-1 or miR-196b has an effect on the morphological structure of muscle fibers in dorsal muscle tissue.

### 3.3. Electron Microscopy Results of Muscle Tissue of the miR-196a-1 and miR-196b Knockout Zebrafish

Transmission electron microscopy was performed on the dorsal muscle tissue of zebrafish, and the experimental results are shown in Figure 4. The longitudinal muscle cells of the dorsal muscle tissue of wild-type zebrafish (Figure 4A) appeared normal in morphology, with a regular arrangement of intracellular myofibrils, normal sarcomere morphology, normal sarcoplasmic reticulum structure, and clear inner ridges of mitochondria. miR-196a-1 gene knockout zebrafish (Figure 4B) showed irregular myocyte morphology, irregular arrangement of intracellular myofibrils, partial myofilament breakage or focal lysis, sarcoplasmic reticulum expansion, mitochondrial swelling, partial inner ridge breakage or outer membrane rupture, and widened myocyte space. The dorsal muscle tissue of miR-196b gene knockout zebrafish (Figure 4C) was longitudinally cut, and it was found that myocytes were irregular in shape, intracellular myofibrils were arranged irregularly, some myofilaments were broken or focally lysed, the sarcoplasmic reticulum was expanded, mitochondria were swollen, some inner ridges were broken or the outer membrane was ruptured, and the myocyte space was widened. In zebrafish with miR-196a-1 and miR-196b gene knockout (Figure 4D), myocytes were irregular in shape, with myofibrils arranged irregularly, Z-lines mostly blurred and disordered, sarcomere swelling, mitochondrial pyknosis, and slightly dilated sarcoplasmic reticulum. The experimental results showed that after miR-196a-1 or miR-196b gene knockout, myofibrils became irregular, and the cell structure was abnormal; that is, muscle tissue was damaged.

### 3.4. Bone CT Detection Results of the miR-196a-1 and miR-196b Knockout Zebrafish Models

Zebrafish with miR-196a-1 and miR-196b gene knockout were collected, and three in each group were tested by micro-CT. The results are shown in Figure 5, and there were no obvious defects in skeletal growth or development in the bones of zebrafish in the experimental groups.

The bone mineral density (BMD) and trabecular number (TB.N) of zebrafish were further analyzed. The head, middle, and tail spines were measured for each fish, and the data obtained are shown in Figure 6.

The BMD of the spine in different parts of the zebrafish was compared (Figure 6A). The BMDs of the head and middle spine of the miR-196a-1 gene knockout group, miR-196b gene knockout group, and miR-196a-1 and miR-196b double knockout group were significantly decreased compared with those of the wild-type zebrafish. The difference between the two groups was statistically significant (*n* = 3, * *p* < 0.05). In the caudal vertebra test results, although the bone mineral density decreased in the gene knockout group, there was no significant difference between the groups. These results suggest that miR-196a-1 or miR-196B gene knockdown reduces spinal BMD or BMD.

In a comparison of the number of trabecular bones in different groups (Figure 6B), it was found that the number of trabecular bones decreased after miR-196a-1 or miR-196b gene knockout. Compared with that of the control group, the number of trabeculae in the middle and tail spine of the miR-196a-1 gene knockout group, miR-196b gene knockout group, and miR-196a-1 and miR-196b double knockout group was significantly decreased, and the difference between the groups was statistically significant (*n* = 3, * *p* < 0.05). In the postcranial vertebra, the number of trabecular bones decreased in the miR-196a-1 gene knockout group, but there was no significant difference between the groups compared with the control group. Compared with that of the single gene knockout group, the number of trabecular bones in the miR-196a-1 and miR-196b double knockout groups was further decreased, but only the middle and tail spine trabecular bone numbers between the miR-196b knockout group and the double knockout group were found to be statistically significant (*n* = 3, * *p* < 0.05), and there was no statistically significant difference between the other groups. The results suggested that miR-196a-1 or miR-196b gene knockdown reduced the number of spinal trabeculae.

In conclusion, miR-196a-1 or miR-196b gene knockout can reduce spine BMD or bone mineral density and trabecular bone number in zebrafish, suggesting that the effect of miR-196a-1 or miR-196b deletion on the zebrafish motor system may be related to the reduction in bone BMD or trabecular bone number.

### 3.5. Real-Time PCR Analysis of Skeletal Muscle and Skeletal-Related Gene Expression Levels

In this study, we screened *vcp*, *dpm1*, *acta1b*, and *mylpfb* genes, which are closely related to the function of skeletal muscle, by using biological information analysis and carried out real-time PCR analysis of zebrafish dorsal muscle tissue. The experimental results are shown in Figure 7. In addition, real-time PCR was used to detect related genes in the bone group, including *col1a1a*, *bmp8a*, *gdf6a*, and *fgfr3*. The experimental results are shown in Figure 8.

Real-time PCR showed that the expression level of the *vcp* gene in the muscle tissue of miR-196a-1 gene knockout zebrafish was significantly lower than that in the control group (*n* = 3, * *p* < 0.05). The expression level of the *vcp* gene in the miR-196b knockout group did not change significantly. In the double-knockout group, the expression level of *vcp* was significantly decreased. Compared with the miR-196a-1 knockout group, there was no significant difference between the groups. Compared with the miR-196b knockout group, the difference between the groups was statistically significant (*n* = 3, ^&^ *p* < 0.05). The expression level of the *dpm1* gene in the miR-196a-1 gene knockout group, miR-196b knockout group, and double knockout group was significantly reduced, and the difference between the groups was statistically significant (*n* = 3, * *p* < 0.05). There was no significant difference in the *dpm1* gene between the miR-196a-1 or miR-196b knockout groups; that is, there was no significant difference between the double knockout group and the single knockout group. Compared with the control group, the expression levels of the *acta1b* and *mylpfb* genes were not significantly different in the miR-196a-1 knockout group. In the miR-196b knockout group, the relative expression levels of the *acta1b* and *mylpfb* genes were significantly reduced, and the difference between the groups was statistically significant (*n* = 3, * *p* < 0.05). In the double knockout group, the expression levels of these genes were also downregulated, and there was no significant difference between the two groups compared with the miR-196b knockout group. Compared with the miR-196a-1 knockout group, the difference between the two groups was statistically significant (*n* = 3, ^#^ *p* < 0.05).

The relative expression level of bone genes was detected. The expression levels of the *gdf6a* and *col1a1a* genes in the miR-196a-1 knockout group, miR-196b knockout group, and double knockout group were significantly lower than those in the control group (*n* = 3, * *p* < 0.05). The relative expression levels of *gdf6a* and *abcc6a* in the double knockout group were lower than those in the miR-196b gene knockout group, and the difference between the two groups was statistically significant (*n* = 3, ^&^ *p* < 0.05). Real-time PCR showed that there was no significant difference in the expression of the *bmp8a* gene in the miR-196a-1 knockout group, the miR-196b knockout group, and the double knockout group; that is, there was no significant difference in the expression of the *bmp8a* gene compared with the control group. The expression level of the *fgfr3* gene in miR-196a-1 knockout zebrafish was significantly decreased, the expression level in the miR-196b knockout group was significantly increased, and the difference between the two groups was statistically significant (*n* = 3, * *p* < 0.05). The expression level of the *fgfr3* gene in the double knockout group was higher than that in the control group, and the difference between the two groups was statistically significant (*n* = 3, ^#^ *p* < 0.05).

## 4. Discussion

Muscle is under the command and control of the nervous system; the force produced by skeletal muscle contraction results in its attached bone movement along with lever movement produced by the joint, resulting in various actions. Bones can support body shape, protect internal organs, maintain body posture, and support weight [26,27]. Skeletal muscle is widely distributed in the body. It plays a contractile function through the reflex of the nervous system and participates in the energy metabolism of the body to reserve energy to meet the needs of exercise [28,29]. Bone and skeletal muscle are closely related and influence each other [30,31]. Skeletal muscle is an important link between bone and bone, provides a microenvironment for bone, and is important in regulating bone latent cells, biomechanics, signaling factors, etc. [32,33]. The skeletal system also plays a role in regulating muscle; for example, paracrine or endocrine factors of osteocytes and osteoblasts can regulate muscle development and muscle strength [34,35]. These two factors complement each other and cooperate to complete the motor function of the body [2]. Dysfunction of bone and muscle leads to a variety of motor system diseases [36,37].

miR-196a plays an important regulatory role in organisms and has been found to be upregulated in a variety of malignant tumors. miR-196a affects the proliferation, invasion, and apoptosis of tumor cells and is expected to become a new molecular marker or target for tumor diagnosis, treatment, and prognosis, indicating its potential in clinical applications [38,39,40,41]. Several studies have also reported that miR-196b is highly expressed in multiple malignant tumors and has the characteristics of an oncogene to promote tumorigenesis [42,43,44]. At present, the relevant studies mainly focus on cancer or tumors, and the roles of miR-196a-1 and miR-196b in bone and skeletal muscle and even in the motor system are still unclear. Studies have found that miR-196a plays an important regulatory role in osteosarcoma, osteoporosis, bone mineral density, and osteoblast differentiation and proliferation [18,20,45]. The relevant reports rarely involve the study of miR-196a, bone and skeletal muscle, or exercise. It was found that pri-miR-196a1 was expressed in the tail and trunk of 24 h zebrafish embryos [21]. Combined with the existing research reports, the results showed that miR-196a plays an important role in bone and skeletal muscle. miR-196b is rarely reported in bone and muscle. Exploring the role of miR-196a and miR-196b in bone or muscle may help elucidate the value of the miR-196 family in the motor system.

Zebrafish are a commonly used animal model for studying bone and muscle. Their bone development has a high degree of similarity to the development process of other vertebrates; their size is small, their feeding cost is low, their reproductive cycle is short, and their reproductive ability is strong [46,47], making them one of the best choices for gene knockout animal models. In this study, miR-196a-1 and miR-196b gene knockout lines were employed. The gene knockout zebrafish lines had no obvious defects in growth and development, indicating that the gene knockout of miR-196a-1 and miR-196b had no significant effect on the overall development of zebrafish and that these embryos could normally develop to maturity. However, whether loss of miR-196a-1 and miR-196b has effects on zebrafish bone and muscle needs to be further studied.

Therefore, we detected the effects of miR-196a-1 and miR-196b on zebrafish motor behavior, as well as the effects on muscle and bone microstructure in the motor system. It was found that the total movement distance, movement duration, speed, and movement times of the zebrafish with miR-196a-1 or miR-196b gene knockout were decreased, especially the related indicators in the double knockout zebrafish, which were further decreased, but there was no significant difference between the miR-196a-1 gene knockout and miR-196b gene knockout groups. The experimental results suggested that the deletion of miR-196a-1 and miR-196b had an impact on motor function, so we further tested the effect of miR-196a-1 and miR-196b gene knockout on the motor system of zebrafish, including muscle and bone.

HE and transmission electron microscopy (SEM) tests of zebrafish dorsal muscle tissue showed that muscle fiber organization of the miR-196a-1 or miR-196b gene knockout zebrafish is irregular, with sarcoplasmic reticulum expansion, mitochondrial swelling, partial spinal fracture or outer membrane rupture, and broadening muscle cell gap observed; namely, the muscle tissue structure was destroyed, and muscle fibers in the double knockout group structure were irregular. These results suggest that the deletion of miR-196a-1 or miR-196b has a certain effect on the morphology and structure of muscle tissue. The bone density of zebrafish spinal bones with CT detection or bone mineral density and bone trabecular number were assessed, and in the knockout group fish, bone density and bone mineral density were reduced, the trabecular bone volume was reduced, no obvious difference between the different knockout groups was found, and the lack of miR-196a-1 or miR-196b influenced the bone density and bone trabecular number. In conclusion, after miR-196a-1 or miR-196b gene knockout, muscle tissue was damaged, bone mineral density was decreased, and the amount of trabecular bone was decreased, which confirmed the effect of gene deletion on motor behavior. We speculated that the miR-196a-1 and miR-196b genes play a role in the motor system by affecting muscle fiber structure, bone mineral density, and the number of trabecular bone.

Valosin-containing protein (*vcp*) has ATP hydrolysis activity and polyubiquitin modification-dependent protein binding activity and is expressed in various tissues, including muscle and brain [48,49]. Some studies have shown that *vcp* is the central mediator of lysosomal clearance and biogenesis in skeletal muscle [50]. Philipp [51] and other researchers showed that deletion of *vcp* in zebrafish damages protein homeostasis and leads to structural and functional defects in striated muscle in vivo. Real-time PCR showed that the expression level of the *vcp* gene in miR-196a-1 knockout zebrafish was significantly decreased, but there was no significant difference in the miR-196b knockout group. Dolichyl-phosphate mannose-transferase subunit 1 (*dpm1*) plays an important role in muscle development and is expected to be active in the endoplasmic reticulum. A study in zebrafish showed that *dpm1* plays an important role in stabilizing muscle structure and regulating apoptotic cells [52]. The expression of the dpm1 gene was downregulated in the knockout group. It is speculated that the stability of muscle structure will be affected after the downregulation of expression, resulting in uneven thickness of muscle fibers in the knockout group. Actin alpha 1 skeletal muscle b (*acta1b*) plays a role upstream or downstream of embryonic cardiac tube development and skeletal muscle fiber development and is expressed in the cardiovascular system and myoblasts [53]. Myosin light chain can phosphorylate fast skeletal muscle b (*mylpfb*), which is expressed in muscle tissue and plays an important role in regulating muscle growth, maintaining muscle structure and function and muscle tissue homeostasis [54]. Real-time PCR showed that the expression levels of the acta1b and *mylpfb* genes were significantly decreased in the miR-196b knockout group and the double knockout group, but there was no significant difference in the miR-196a-1 knockout group, suggesting that after miR-196b gene knockout, the morphological structure and function of muscle might be affected by the acta1b and mylpfb genes. The differential expression of these genes may be related to the structural and functional changes in muscle.

The bone extracellular matrix plays an important role in the dynamic action of osteoblasts and osteoclasts to regulate the process of bone regeneration. Col1a1 is an important part of the bone matrix [55]. Changes in the composition of the bone extracellular matrix can destroy ECM-bone cell signal transduction, resulting in changes in bone mineral density and/or bone microstructure. The expression level of *col1a1a* was significantly downregulated after miR-196a-1 or miR-196b gene knockout. In combination with previous experimental results, it was found that the bone mineral density and the number of bone trabeculae were decreased in zebrafish after miR-196a-1 and miR-196b gene deletion. It is speculated that the deletion of the miR-196a-1 and miR-196b genes may affect the expression of the collagen gene or affect bone mineral density and the number of bone trabeculae through signal transduction, thus affecting the function of zebrafish bone. Bone morphogenetic protein 8a (*bmp8a*) is a member of the bone morphogenetic protein family, which is a classical multifunctional growth factor and belongs to the transforming growth factor β superfamily, and *bmp8a* plays an important regulatory role in bone, muscle, blood vessels, and other tissues [56]. In this study, it was found that there was no significant difference in the expression of the *bmp8a* gene in the miR-196a-1 or miR-196b gene knockout group. It is speculated that the effect of miR-196a-1/b on bone may be unrelated to bmp8a. Growth differentiation factor 6a (*gdf6a*) is expressed in a variety of tissues and structures. It is a ligand of the bone morphogenetic protein family that is expressed in the notochord and primitive intestinal endoderm of zebrafish. The human homologous gene of this gene is related to a variety of diseases, including multiple joint fracture syndrome [57]. The expression level of the *gdf6a* gene was significantly downregulated in the miR-196a-1 or miR-196b gene knockout group and was further downregulated in the double knockout group. Fibroblast growth factor receptor 3 (*fgfr3*) is expressed in the chondrocytes of zebrafish head cartilage, osteoblasts involved in bone formation, and other cells [58]. In this study, we found that the expression level of *fgfr3* was downregulated in the miR-196a-1 gene knockout group and upregulated in the miR-196b gene knockout group. In the double knockout group, the relative expression level of the fgfr3 gene was higher than that in the miR-196a-1 knockout group and lower than that in the miR-196b knockout group.

In summary, miR-196a-1 or miR-196b knockout had a certain impact on the bone mineral density and trabecular quantity of zebrafish bone tissue, and there was no significant difference in the changes in bone microstructure between the double knockout group and the single knockout group. It is speculated that in the miR-196a-1 or miR-196b knockout group, these changes in bone microstructure may be related to the changes in the expression levels of *gdf6a*, *fgfr3*, and *col1a1a*.

Our results show that the effects of miR-196a-1 and miR-196b on bone and muscle may be related to the expression levels of *vcp*, *dpm1*, *acta1b*, *mylpfb*, *gdf6a*, *fgfr3*, and *col1a1a*. We expect to study the key factors associated with miR-196a-1 and miR-196b in muscle and bone to further elucidate the mechanism of miR-196a-1 and miR-196b in organisms.

## 5. Conclusions

The motor function of zebrafish decreased after miR-196a-1 or miR-196b gene knockout. Microstructure analysis showed that zebrafish with gene knockout had abnormal muscle fiber structure, reduced bone mineral density, and reduced trabecular bone data. We hypothesized that miR-196a-1 or miR-196b causes motor function decline by affecting muscle and bone tissue, and that these effects may be related to the expression of the *vcp*, *dpm1*, *acta1b*, *mylpfb*, *gdf6a*, *fgfr3*, and *col1a1a* genes. These results suggest that miR-196a-1 and miR-196b play roles in muscle fiber structure, bone mineral density, and bone trabecular quantity by affecting the expression of *vcp*, *dpm1*, *acta1b*, *mylpfb*, *gdf6a*, *fgfr3*, and *col1a1a*, and then affect the function of the motor system.

## Figures and Tables

**Figure 1 biomolecules-13-00554-f001:**
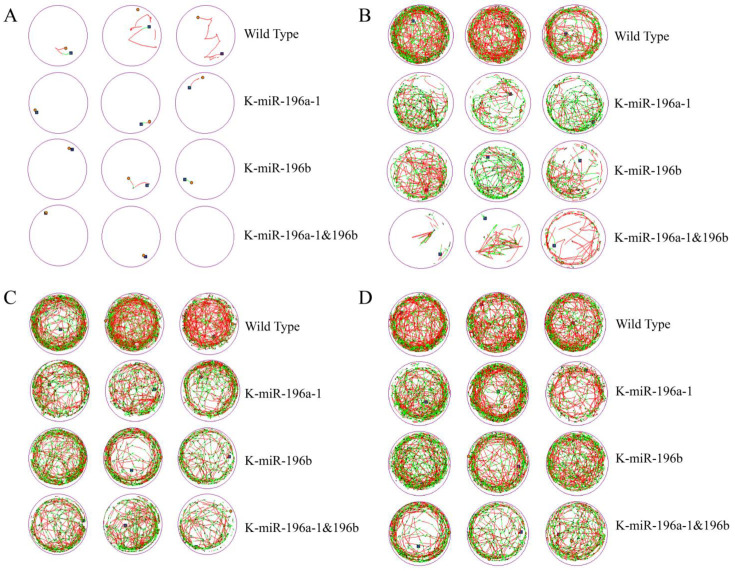
Behavioral trajectory of miR-196a-1 and miR-196b gene knockout zebrafish. The mobile trajectory was observably reduced in the miR-196a-1 or miR-196b gene knockout zebrafish group. (**A**): Mobile trajectory of zebrafish on the fourth day; (**B**): mobile trajectory on the fifth day; (**C**): mobile trajectory on the sixth day; (**D**): mobile trajectory diagram on the seventh day. WT, wild-type zebrafish; K-miR-196a-1, miR-196a-1 gene knockout zebrafish; K-miR-196b, miR-196b gene knockout zebrafish; K-miR-196a-1/196b, zebrafish with miR-196a-1 and miR-196b gene knockout. The blue square is the starting point, the orange dot is the ending point, the red and green lines are the different speeds of trajectories, the green relative speed is medium, and the red relative speed is higher.

**Figure 2 biomolecules-13-00554-f002:**
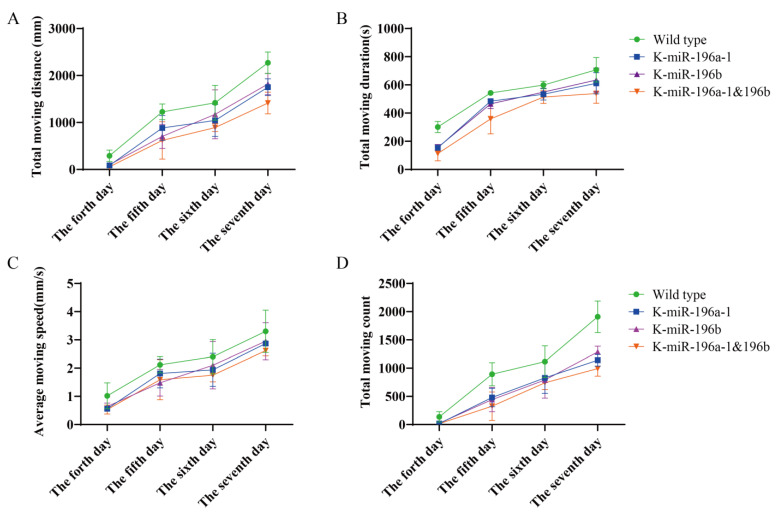
Analysis results of behavioral trajectory data of different groups of zebrafish. The green line indicates wild-type zebrafish; the blue line represents the miR-196a-1 gene knockout group (K-miR-196a-1); and the purple line represents the zebrafish of the miR-196b gene knockout group (K-miR-196b). Orange is the behavior of the miR-196a-1 and the miR-196b double mutant (K-miR-196a-1/196b). (**A**): The total movement distance of zebrafish (mm) was reduced in the gene knockout group, especially in the double knockout group; (**B**): the total movement time of zebrafish was decreased after knockout of miR-196a-1 and miR-196b; (**C**): the movement speed (mm/s) was decreased in the gene knockout group; (**D**): the movement times of the gene knockout zebrafish were decreased compared with those of the wild-type, but there were no obvious differences between the miR-196a-1 and miR-196b knockout groups.

**Figure 3 biomolecules-13-00554-f003:**
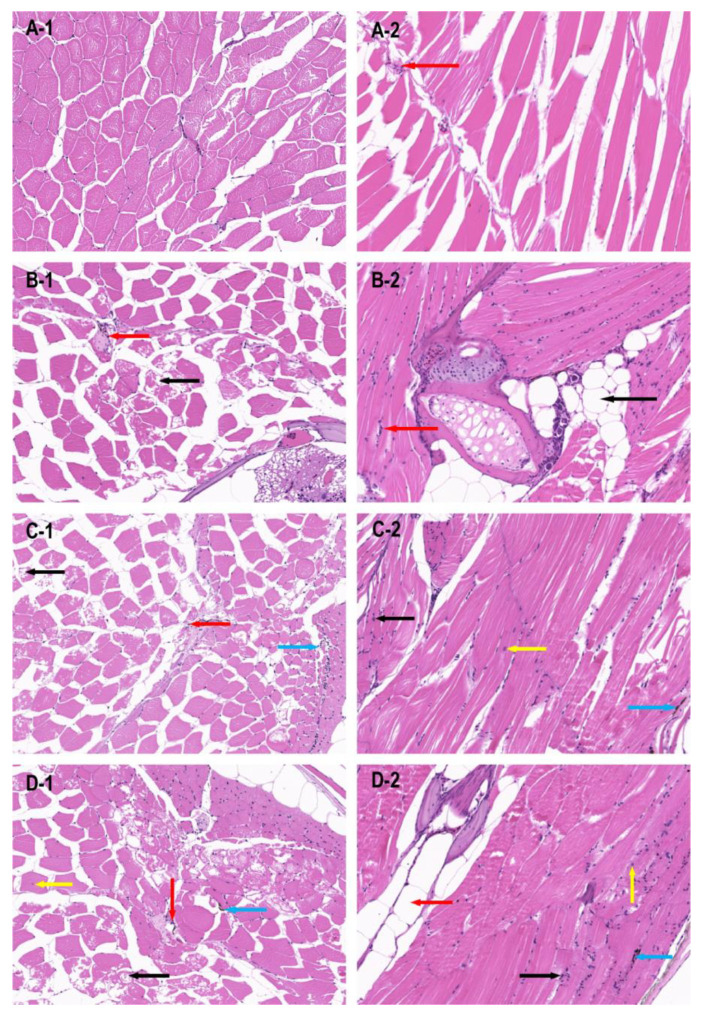
HE staining results of dorsal muscle tissue of zebrafish in different groups (20×).The dorsal muscle tissue was damaged when the miR-196a-1 and miR-196b genes were knocked out. (**A**) wild-type zebrafish; (**B**): zebrafish with miR-196a-1 gene knockout; (**C**): miR-196b gene knockout zebrafish; (**D**): zebrafish with miR-196a-1 and miR-196b gene knockout; -1 is the transverse section of muscle tissue, -2 is the longitudinal section of muscle tissue. The red arrow points to inflammatory cell infiltration, the black arrow points to vacuolar degeneration of muscle fibers, the blue arrow points to melanin deposition, the yellow arrow points to inward nuclear movement.

**Figure 4 biomolecules-13-00554-f004:**
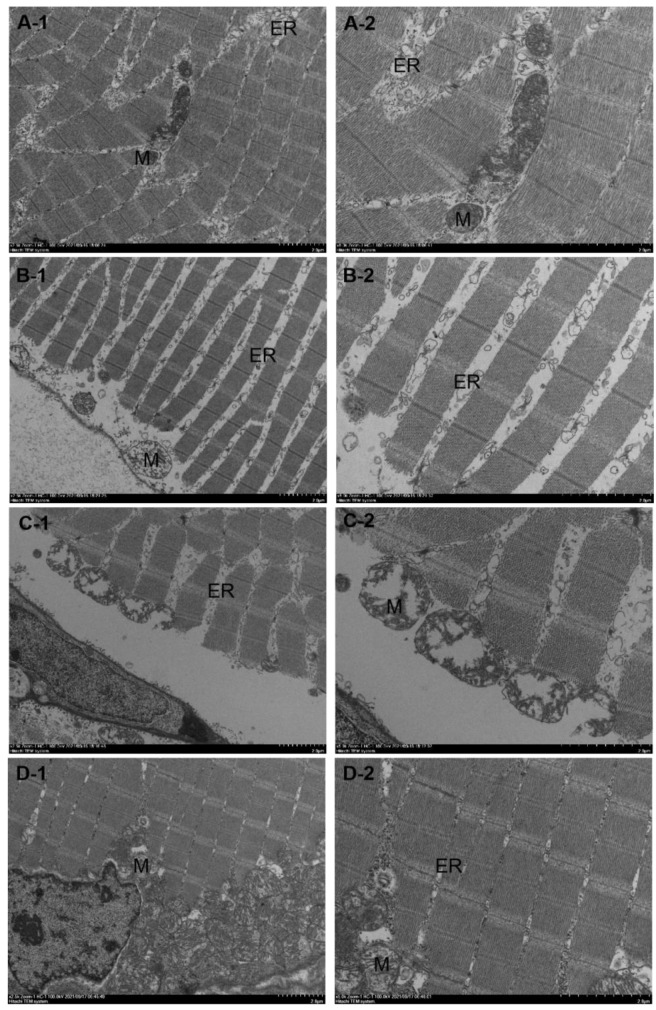
Transmission electron microscopy results of dorsal muscle tissue of miR-196a-1 and miR-196b knockout zebrafish. The zebrafish muscle tissue was damaged after miR-196a-1 or miR-196b gene knockout and mainly showed irregular morphology of muscle cells, irregular arrangement of myofibrils, sarcoplasmic reticulum dilatation, and mitochondrial swelling. (**A**): Wild-type zebrafish; (**B**): zebrafish with miR-196a-1 gene knockout; (**C**): miR-196b gene knockout zebrafish; (**D**): zebrafish with miR-196a-1 and miR-196b gene knockout; left panels: 2500×, right panels: 5000×. ER is sarcoplasmic reticulum, M is mitochondria, and N is muscle nucleus.

**Figure 5 biomolecules-13-00554-f005:**
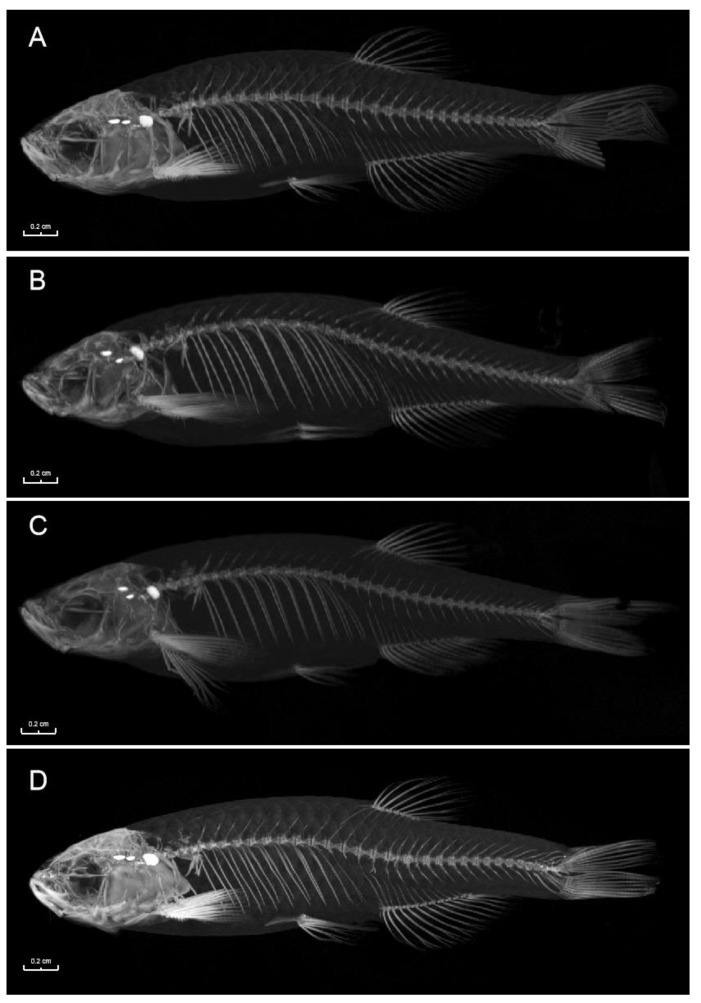
CT scanning results of zebrafish bones of different groups under CT. The white part is the zebrafish skeleton. No obvious difference was found in the bone structure of the miR-196a-1 and miR-196b gene knockout zebrafish. (**A**): Wild-type zebrafish; (**B**): zebrafish with miR-196a-1 gene knockout; (**C**): miR-196b gene knockout zebrafish; (**D**): zebrafish with miR-196a-1 and miR-196b gene knockout.

**Figure 6 biomolecules-13-00554-f006:**
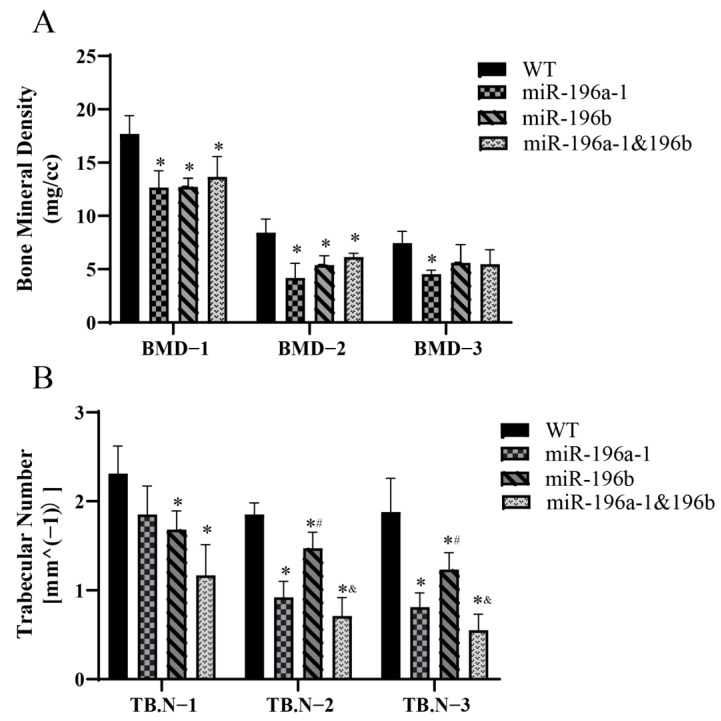
Results of bone mineral density and trabecular number in zebrafish knockout models. (**A**) The mineral density and (**B)** trabecular number were decreased in the miR-196a-1 or miR-196b knockout group. BMD-1 is the BMD or bone mineral density of the postcranial vertebra, BMD-2 is the middle spinal segment, and BMD-3 is the caudal vertebra. TB.N-1 is the trabecular number of the postcranial vertebra, TB.N-2 is the middle spinal segment, and TB.N-3 is the caudal vertebra. WT is wild-type zebrafish. K-miR-196a-1 is the miR-196a-1 gene knockout zebrafish; K-miR-196b is the miR-196b gene knockout zebrafish; K-miR-196a-1/196b is the zebrafish with miR-196a-1 and miR-196b gene knockout; compared with wild zebrafish, * *p* < 0.05; compared with miR-196a-1 gene knockout zebrafish, ^#^
*p* < 0.05; compared with miR-196b gene knockout group, ^&^
*p* < 0.05, *n* = 3.

**Figure 7 biomolecules-13-00554-f007:**
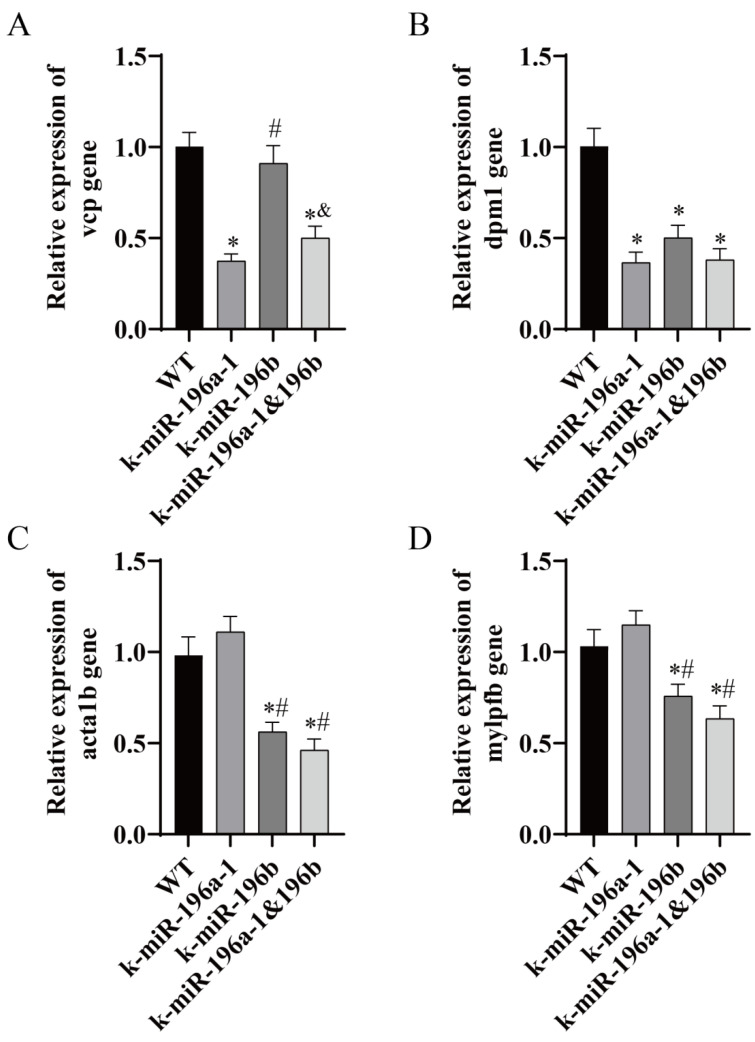
The expression of muscle-related genes in miR-196a-1 or miR-196b knockout zebrafish was determined by real-time PCR. WT indicates wild-type zebrafish; K-miR-196a-1 indicates miR-196a-1 gene knockout zebrafish; K-miR-196b indicates miR-196b gene knockout zebrafish; K-miR-196a-1/196b indicates miR-196a-1 and miR-196b gene knockout zebrafish. (**A**) shows the relative expression level of the *vcp* gene, (**B**) is the *dpm1* gene; (**C**) is the *acta1b* gene; (**D**) is the *mylpfb* gene. Compared with wild-type zebrafish, * *p* < 0.05; compared with miR-196a-1 gene knockout zebrafish, ^#^
*p* < 0.05; compared with miR-196b gene knockout zebrafish, ^&^
*p* < 0.05, *n* = 3.

**Figure 8 biomolecules-13-00554-f008:**
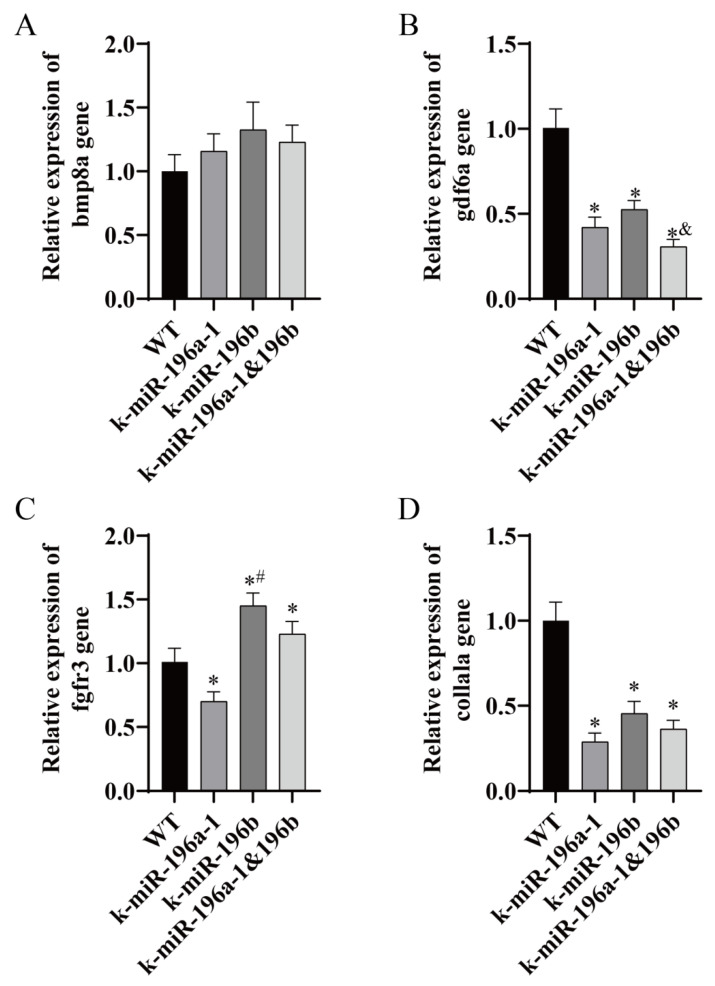
The expression of bone-related genes in miR-196a-1 or miR-196b knockout zebrafish was determined by real-time PCR. WT indicates wild-type zebrafish; K-miR-196a-1 indicates miR-196a-1 gene knockout zebrafish; K-miR-196b indicates miR-196b gene knockout zebrafish; K-miR-196a-1/196b indicates miR-196a-1 and miR-196b gene knockout zebrafish. (**A**) shows the relative expression level of the *bmp8a* gene, (**B**) is the *gdf6a* gene; (**C**) is the *fgfr3* gene; (**D**) is the *col1a1a* gene. Compared with wild-type zebrafish, * *p* < 0.05; compared with miR-196a-1 gene knockout zebrafish, ^#^
*p* < 0.05; compared with miR-196b gene knockout zebrafish, ^&^
*p* < 0.05, *n* = 3.

## Data Availability

The data are available from the corresponding author.

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
