# Peer review of "Roles of miR-196a and miR-196b in Zebrafish Motor Function"

_biomolecules, 2023, doi:10.3390/biom13030554_

Round 1

Reviewer 1 Report

Yuan et al., investigate the role of miR-196a and miR-196b in zebrafish motor function by producing corresponding knockout fish. They investigate their locomotor behaviour, bone and muscle maturation, and the expression of relevant genes for bone and muscle function.

Although the motility results and the observed decrease in mineral density and trabecula count are very interesting, the study does have some weaknesses.

1. The reviewer misses the evidence of the knockout status of the corresponding miRNas in the fish.

2. why are differences in muscle and bone anatomy only examined in 10-month-old zebrafish?

3. the discussed differences in muscle anatomy between WT and knockout fish are not apparent to the reviewer in the figure 3 shown.

4. the same applies to the bone anatomy in figure 4.

5. When examining the expression of individual genes that the authors see influenced by miR-196a and miR-196b, it is not understood whether the authors analyze separated muscle and bone tissue. Again, it is not understood why 10-month-old fish (they are probably not 10 years old as described in line 110) are being studied, whereas the altered motility of knockout newborns should have immediate physiological and molecular effects.

Author Response

Dear Expert, Thank you very much for your recognition of our research and your constructive and valuable comments on our manuscript. We agree with your suggestions and have made the appropriate changes in the paper based on the comments. We believe the quality of this manuscript has been significantly improved with your help and guidance. Thank you very much. We wish you all the best in your work and good health!

Our response is as follows.

  1. The reviewer misses the evidence of the knockout status of the corresponding miRNas in the fish.

Response: Dear experts, thank you for your valuable comments. We have successfully constructed models of miR-196a and miR-196b knockout in zebrafish, and the methods and evidence of knockout status have been published in the corresponding Chinese literature. (https://kns.cnki.net/kcms2/article/abstract?v=3uoqIhG8C44YLTlOAiTRKibYlV5Vjs7i8oRR1PAr7RxjuAJk4dHXonMoI40Dse8VTGC7_c32dCHRA0ZpbLuYJaSb73WL7pKl&uniplatform=NZKPT)Our previous series of studies found that this genotype of zebrafish has become a model capable of mature and stable breeding, so we have not placed evidence of knockout status in this article. Our pre-knockout status evidence is shown in Fig.1, Fig.2 and Fig.3 below.

Fig.1 Analysis of effectiveness of embryo injection  Gel electrophoresis of PCR products for embryo injection effectiveness analysis. M: DNA marker; 1~5: PCR amplification products of genomic DNA with embryo injection; 6: Wild type control group. In addition to a bright 210 bp wild-type band in lanes 3 and 4,there is a weaker band ( indicated by a black arrow) below,indicating that both target sites are valid

Fig.2 F0 generation two months juvenile screening  M: DNA marker; 1~9: PCR amplified products using two months juvenile genomic DNA injected with miï¼²-196 guide ï¼²NAs; 10: Wild type control. In addition to a bright 210 bp wildtype band,lane 8 has a weaker band below ( Indicated by the black arrow)

Fig.3 F1 generation mutant screening (a) Gel electrophoresis of PCR products using F1 embryos as genomic DNA. M: DNA marker; 1~9: PCR products using genomic DNA injected with sgï¼²NA; 10: Wild type control. In addition to a bright 210 bp wild-type band,there was a weaker band below in lanes 1,3,and 7 ( indicated by the black arrow) . The fish No. 8 selected by the F0 mutant can produce mutant offspring that can be stably inherited; (b) The F1 generation mutant embryo PCR product sequencing peak map,the small band of PCR product was purified and sent to the company for sequencing,“———”indicated the first target site sequence,7 bases missing from target site 1; “……”marked the second target site sequence,7 bases missing at target 2; (c) Mutant sequence was compared with the control in the NCBI blast datebase,103 bp was missing between the first and second target sites; (d) Gel electrophoresis of PCR products of F1 adult mutant fish. M: DNA marker; 1~23: PCR amplified product of genomic DNA with embryo injection; 24: Wildtype control.Inaddition to a bright 210 bp wildtype band,there was a weaker band below in lanes 1,3,10,12,19 ( indicated by the black arrow).

  1. why are differences in muscle and bone anatomy only examined in 10-month-old zebrafish?

Reply: Hello, Dear expert, thank you for your valuable comments. We used 10-month-old zebrafish for the study for the following two reasons: on the one hand, to observe whether there is a certain mortality rate in zebrafish growing to adulthood after miR-196a and miR-196b gene knockout, but actually no mortality was observed; on the other hand, it is hoped that zebrafish grow to a size that is conducive to the separation of skeletal muscle and bone, and the appropriate size is also conducive to carrying out micro-CT scan, and the subsequent calculation of bone density. From the above considerations, we chose zebrafish of 10 months of age. Thank you.

  1. the discussed differences in muscle anatomy between WT and knockout fish are not apparent to the reviewer in the figure 3 shown.

Response: Dear expert, thank you for your valuable comments. We used a microscope with magnification of 200x to obtain a larger field of view, which resulted in the pathological structures in the figure being slightly unclear after the images were grouped, and we have replaced the images. Thank you.

  1. the same applies to the bone anatomy in figure 4.

Reply: Hello respected experts, thank you for your valuable comments. We have adjusted the pictures accordingly.

  1. When examining the expression of individual genes that the authors see influenced by miR-196a and miR-196b, it is not understood whether the authors analyze separated muscle and bone tissue. Again, it is not understood why 10-month-old fish (they are probably not 10 years old as described in line 110) are being studied, whereas the altered motility of knockout newborns should have immediate physiological and molecular effects.

Response: Hi Dear expert, thank you for your valuable comments. When we performed gene expression assays on miR-196a and miR-196b knockout zebrafish, we isolated muscle tissue and bone tissue to explore gene expression in both tissues. We observed the motility of knockout zebrafish juveniles, but not adults, mainly because we found that the autonomous ability of juveniles is easy to observe and count and more accurate, while the autonomous motility of adults is difficult to observe and count and is not accurate. We successively observed the movement trajectories of zebrafish juveniles at the age of day four, five, six and seven, and all of them were found to show diminished motility in miR-196a or miR-196b knockout zebrafish.

Reviewer 2 Report

This is an interesting work with relatively novel approach, however, there are some minor comments below need to be addressed;

- There are some minor punctuation mistakes, so please provide a detailed proof read.

- In the introduction and the abstract, please clearly emphasize the novelty of this work.

- Please justify clearly why the zebrafish model was selected.

- Please provide more details about your experimental section and statistical analyses.

- The results seem consistent, but looks like the combined knockout of both miRNA has similar effect of single miRNA for most cases either dominated by miR196a-1 or miR196b depending on the function. This dampens the enthusiasm about the hypothesis. Rather than combined use, discussing the individual effect in more detail might help, which was done in the discussion section to a certain extend.

Author Response

Dear Expert, Thank you very much for your recognition of our research and your constructive and valuable comments on our manuscript. We agree with your suggestions and have made the appropriate changes in the paper based on the comments. We believe the quality of this manuscript has been significantly improved with your help and guidance. Thank you very much. We wish you all the best in your work and good health!

Our response is as follows.

1.There are some minor punctuation mistakes, so please provide a detailed proof read.

Reply: Dear experts, thank you for your valuable comments. After our careful reading, we did find that there were punctuation problems in the text. We are very sorry for such a low-level error in the revision process, and we have made changes and marked it as a red word, thank you.

2.In the introduction and the abstract, please clearly emphasize the novelty of this work.

Response: Dear experts, thank you for your valuable comments. The role of miR-196a-1 and miR-196b in skeletal and skeletal muscle is still unclear. In order to explore the role of miR-196 family in skeletal and skeletal muscle, we successfully constructed miR-196a-1 and miR-196b knockout zebrafish models. We also performed preliminary tests on the motility function of miR-196 knockout zebrafish, and found that miR-196 gene loss in zebrafish decreased motility function, suggesting that miR-196 may have an effect on the performance of motility function in zebrafish, but the related mechanism of action is not clear. Therefore, to investigate the effects and possible mechanisms of effects of miR-196 on skeletal and skeletal muscle and related motility functions, we performed a series of experiments on miR-196a-1 and miR-196b knockout zebrafish related to functional assays, behavioral and motility function assays, tissue structure assays, and molecular biology studies to explore the effects of miR-196a-1 and miR 196b on skeleton and skeletal muscle. The results will provide a theoretical basis for the prevention and treatment of skeletal and skeletal muscle-related diseases and improvement of motility in clinical applications.

We have added novelty to the introduction and abstract and marked it as red words, thanking you for your valuable comments.

3.Please justify clearly why the zebrafish model was selected.

Response: Dear experts, thank you for your valuable comments. The reasons for the choice of zebrafish were explained in our original manuscript as follows:“Zebrafish is a commonly used animal model for studying bone and muscle. Its bone development has a high degree of similarity to the development process of other vertebrates; its size is small, its feeding cost is low, its reproductive cycle is short, and its reproductive ability is strong [43, 44], making it one of the best choices for gene knockout animal models.”

Furthermore, in zebrafish, skeletal muscle and bone make up a large part of the body trunk and are highly similar to human muscle both molecularly and histologically, making them suitable for the study of bone and muscle diseases [1]. And the relatively simple genome and simple genetic manipulation of zebrafish are advantageous in myopathy studies [2].

We have added to the introduction and marked the word in red, thank you for your valuable comments.

[1]Daya A, Donaka R, Karasik D. Zebrafish models of sarcopenia. Dis Model Mech. 2020;13(3):dmm042689. Published 2020 Mar 30. doi:10.1242/dmm.042689

[2]Gupta V, Kawahara G, Gundry SR, et al. The zebrafish dag1 mutant: a novel genetic model for dystroglycanopathies. Hum Mol Genet. 2011;20(9):1712-1725. doi:10.1093/hmg/ddr047

4.Please provide more details about your experimental section and statistical analyses.

Reply: Hello, Dear experts, thank you for your valuable comments. We have made supplements in the text for the experimental part and statistical analysis and marked it in red words, thank you for your valuable comments.

5.The results seem consistent, but looks like the combined knockout of both miRNA has similar effect of single miRNA for most cases either dominated by miR196a-1 or miR196b depending on the function. This dampens the enthusiasm about the hypothesis. Rather than combined use, discussing the individual effect in more detail might help, which was done in the discussion section to a certain extend.

Response: Dear experts, thank you for your valuable comments. We constructed miR196a-1 knockout, miR196b knockout and miR196a-1, miR196b double knockout zebrafish models and found that the gene knockout zebrafish lines had no obvious defects in growth and development, indicating that the gene knockout of miR-196a-1 and miR-196b had no significant effect on the overall development of zebrafish and that these embryos could normally develop to maturity.The motor function of zebrafish decreased after miR-196a-1 or miR-196b gene knockout. Microstructure analysis showed that zebrafish with gene knockout had abnormal muscle fiber structure, reduced bone mineral density, and reduced trabecular bone data. We hypothesized that miR-196a-1 or miR-196b causes motor function decline by affecting muscle and bone tissue and that these effects may be related to the expression of the vcp, dpm1, acta1b, mylpfb, gdf6a, fgfr3, and col1a1a genes. These results suggest that miR-196a-1 and miR-196b play roles in muscle fiber structure, bone mineral density and bone trabecular quantity by affecting the expression of vcp, dpm1, acta1b, mylpfb, gdf6a, fgfr3, and col1a1a and then affect the function of the motor system. We will further explore the individual effects of miR196a-1 and miR196b in the future.

Round 2

Reviewer 1 Report

no further comments